# AskMe: A LAPPS Grid-based NLP Query and Retrieval System for Covid-19 Literature

**Keith Suderman[1], Nancy Ide[1], Marc Verhagen[2], Brent Cochran[3], James Pustejovsky[2]**
[1]Vassar College [2]Brandeis University [3]Tufts University

{suderman,ide}@cs.vassar.edu, {marc,jamesp}@cs.brandeis.edu, brent.cochran@tufts.edu

## Abstract

In a recent project, the Language Applications Grid was augmented to support the mining of scientific publications. The results of that effort have now been repurposed to focus on Covid-19 literature, including modification of the LAPPS Grid "AskMe" query and retrieval engine. We describe the AskMe system and discuss its functionality as compared to other query engines available to search covid-related publications.

## 1 Introduction

The onset of the coronavirus pandemic has prompted concerted efforts within the field of natural language processing (NLP) to enable researchers and practitioners to easily access and mine the growing body of literature concerned with the virus. Between February and May 2020, the number of scientific papers published on COVID-19 research increased from 29,000 to more than 138,000; this number is expected to exceed one million by the end of 2020.

In a recent project, the Language Applications (LAPPS) Grid[1] (Ide et al., 2014a) was augmented to support the mining of scientific publications[2] (Ide et al., 2018). The results of that effort have now been repurposed to focus on Covid-19 literature, including modification of the LAPPS Grid "AskMe" query and retrieval engine to access nightly updates of the CORD-19 dataset[3] (Wang et al., 2020) available from the Allen Institute for AI.

In this paper, we describe the AskMe system and discuss its functionality as compared to other query engines available to search covid-related publications. Because the AskMe engine is deployed primarily as a front end to the LAPPS Grid, one of its most salient features is the ability to further process query results with the large array of NLP tools available in the Grid. The AskMe system also provides numerous options for displaying results as well as means to fine-tune parameters of the search heuristics to suit particular needs and interests.

## 2 Related Work

Several literature search engines have been developed or adapted in the wake of the Covid-19 pandemic in order to expedite the progress of research focusing on the virus. Among these are facilities developed by major corporations, most notably IBM (IBM Inc., 2020), Amazon Web Services (AWS, 2020), and Google (Google Inc., 2020), which generally do not access daily updates of Covid-19 literature and may include licensed databases.

Non-commercially developed search engines for Covid-19 data include Vespa, LitCovid and iSearch. Vespa (Vespa, 2020) provides an open source search engine for version 2020-05-19 of the CORD-19 dataset. LitCovid (Chen et al., 2020) is a curated open-resource literature hub developed by the National Institutes of Health (NIH) and the National Library of Medicine (NLM), and is arguably the most comprehensive freely-available resource for Covid-19 literature discovery. Its database is limited to articles in PubMed specific to COVID-19, vs. CORD-19, which includes other viruses (e.g., SARS, MERS) and covers a time period well before the current outbreak. LitCovid divides the articles into categories (e.g. Mechanism, Transmission, Diagnosis, and Treatment), and, based on automatically generated annotations, reports the chemicals, journals, and countries contained in the results. Results can be sorted by relevance or recency.

NIH's iSearch COVID-19 portfolio (National

---

[1]https://galaxy.lappsgrid.org

[2]Funded by U.S. National Science Foundation grant NSF-EAGER 181123.

[3]https://www.kaggle.com/allen-institute-for-ai/CORD-19-research-challenge/data

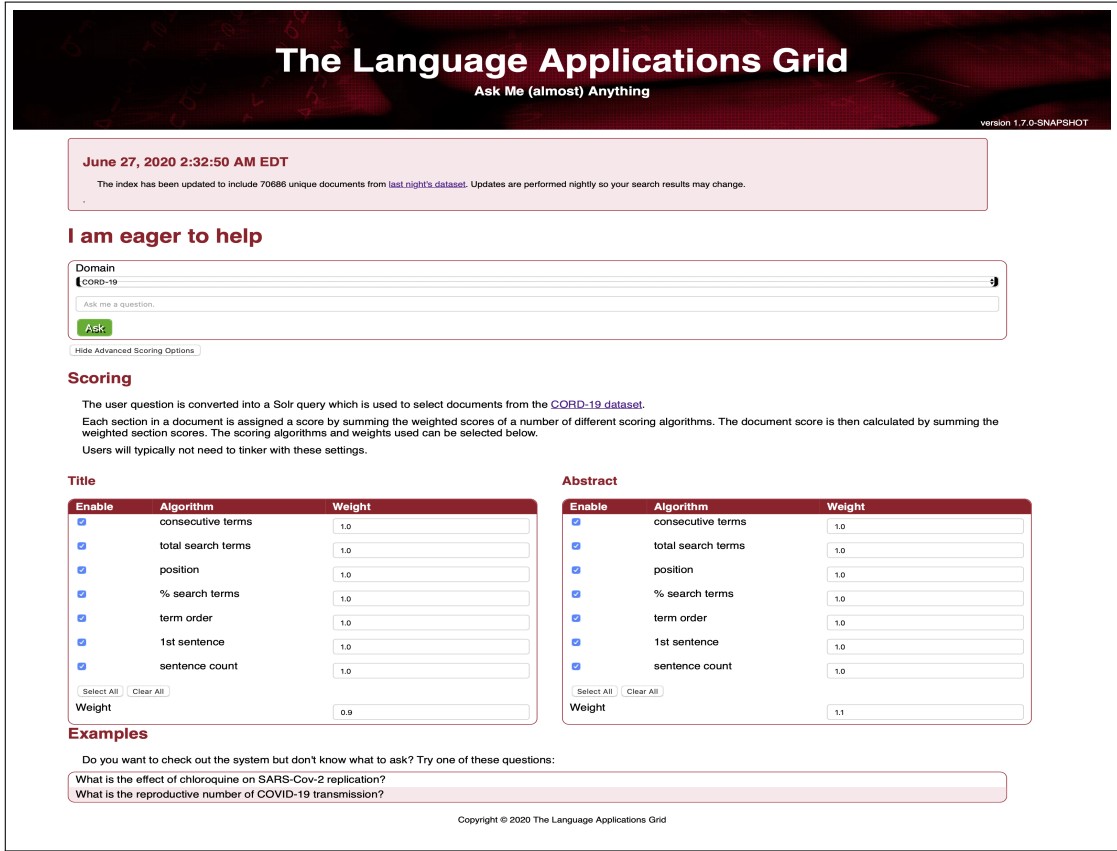

Figure 1: AskMe Query Interface

Institutes of Health, 2020), also updated daily, includes both publications and preprints (from medRxiv, SSRN, arXiv, bioRxiv, Research Square and ChemRxiv) and provides interactive visualizations that allow users to select topics within their search results for download. The user can view results as a list in either chronological or reverse chronological order, or in "facet view" which shows results grouped by category (date, journal, author, etc.). The system provides advanced filters for more exact timeframes, publications types, source dataset, and inclusion or exclusion of iCite articles.

## 3 LAPPS Grid

The LAPPS Grid provides a large collection of NLP tools from a wide variety of sources exposed as web services, together with multiple commonly used resources (e.g., gold standard corpora). The services and resources are made available via the Galaxy web-based workflow development engine (Afgan et al., 2018), and may also be accessed directly via SOAP calls or programmatically through Java and Python interfaces. All tools and resources in the LAPPS Grid are rendered mutually interop-

erable via transduction to the LAPPS Grid Interchange Format (LIF) (Verhagen et al., 2015) and the Web Service Exchange Vocabulary (WSEV) (Ide et al., 2014b), both designed to capture fundamental properties of existing annotation models in order to serve as a common pivot among them. The LAPPS Grid also provides interoperable, two-way access to the PubAnnotation annotation repository (Kim and Wang, 2012), the INCEpTION machine learning-assisted annotation platform (Klie et al., 2018), along with multi-lingual NLP tools and data in two EU-CLARIN platforms: WebLicht (Hinrichs et al., 2010) and LINDAT/CLARIN (Straka et al., 2016). All LAPPS Grid components are released under the Apache 2.0 open source license.

## 4 AskMe Overview

The AskMe application is an open-source, Solr-based microservice architecture running in a Docker Swarm on Jetstream (Stewart et al., 2015; Towns et al., 2014), an NSF-funded compute cluster. Originally developed to allow for search and retrieval from the PubMed database, it now allows for search of the full set of CORD-19 documents, which are updated nightly.

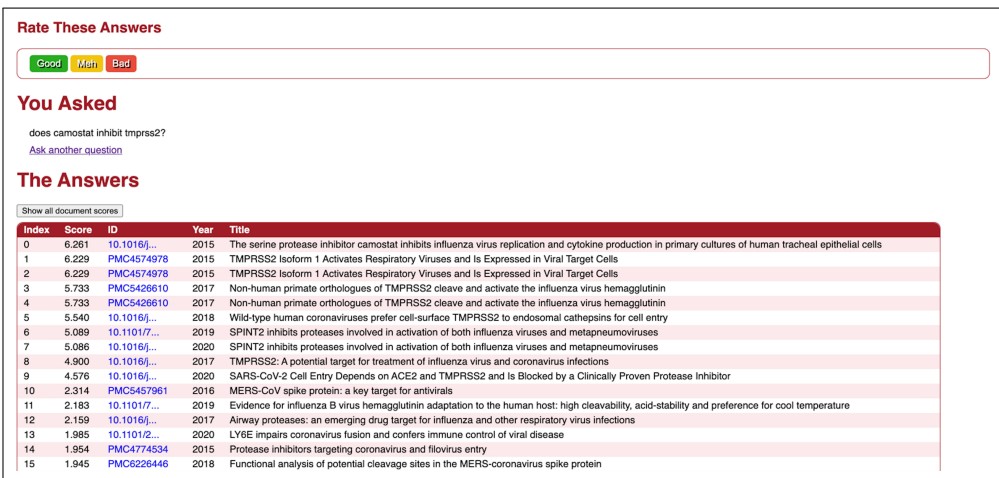

Figure 2: AskMe Query Results

In response to a natural language query, AskMe applies several scoring algorithms to hone the results of Solr's tf-idf similarity processing (based on the vector space model). Currently, it scores only a document's title and abstract; AskMe will be expanded to handle full text in the near future. The score for each of these sections is computed from the following:

1. The total number of search terms that appear, including duplicates–e.g., if the seach terms are *X Y* and the section contains *...X...Y...X...* (where periods represent tokens that are not search terms), the score is 0.2 (3/15).

2. The percentage of the search terms that appear. In the above example the score would be 1.0 since all search terms appear in the section.

3. The order of terms in the search query, e.g., if the search terms are *X Y* then *...X...Y...* will score higher than *...Y...X...*.

4. The total number of sentences that contain one or more search terms.

5. The percentage of search terms that appear in the first sentence.

6. Sequences of consecutive of search terms, e.g., if the search terms are *X Y* and the section contains *...X.YYX..XY* the score would be 0.4167 (5/12) as there are two sequences of consecutive terms (*YYX* and *XY*, with an aggregate length of 5) in the 12 tokens.

A section's score is the sum of the component algorithm scores; the overall document score is computed by adding the scores for the individual sections. The main AskMe page (Figure 1) includes an option to adjust the weights for inividual scoring algorithms to suit specific document types or interests, or simply fine-tune results. Results can be viewed as a list including only score and title (Figure 2) or coupled with document scores for each of the component algorithms. In the latter view, overall score for each section is highlighted. Clicking on any column in either view sorts the results according to the values for that column. The full text for any document in the results can be viewed by clicking on the document ID. Once results are generated, the user may directly import them into the LAPPS Grid for processing by any of a wide array of NLP tools from popular sources (Stanford, OpenNLP, LingPipe, and many others), for example to identify entities, relations, dependencies, etc. Annotations can be manually edited by accessing the editor in either PubAnnotation or INCEpTION and brought back into the Grid for further processing, e.g., to train any of the various machine learning algorithms available in the LAPPS Grid.[4]

## 5 Comparison with Other Systems

We limit our comparison to LitCovid and iSearch, the most popular Covid literature search engines. In terms of functionality, LitCovid and iSearch currently provide more advanced filtering by date, source, journal, etc. than AskMe, although there are plans to add such options (see Section 6). All provide one-click access to the original publications. The most salient differences between AskMe and other systems include:

- Access to the latest version of the CORD-19

---

[4]See (Eckart de Castilho et al., 2019) for a description of the interactive use of the three systems.

dataset, which includes a larger set of documents than the curated LitCovid and iSearch.

- Ability to sort on any column, including composite and component scores. LitCovid allows sorting on relevance and recency only, while iSearch allows sorting on any column in list view (title, journal, source, etc.). Neither LitCovid nor iSearch provide scores; relevance is determined by sorted display order.
- Elimination of duplicates. It is often the case that an article is found in more than one of the several sources in the corpus, LitCovid therefore includes duplicates in the results.
- Ability to limit the number of results to return.
- Ability to see as well as fine-tune components of the scoring algorithm.
- Ability to export results for annotation by LAPPS Grid NLP tools.
- Ability to export annotations for contribution to PubAnnotation, edit annotations in INCEp-TION or PubAnnotation's TextEdit, and re-import results from either platform for additional processing if desired.

Comparison of AskMe results with those of other query engines is not straightforward, as none of the systems described in Section 2 works exclusively with the most recently (daily) updated version of CORD-19. Therefore, we first focus on comparing AskMe and LitCovid, which is the most closely related and relevant system, and include only documents included in both the LitCovid and CORD-19 corpora (as of 28 June 30 2020, 16,929 documents) in the results. Table 1 shows the top ten results from LitCovid in response to the query "What is the effect of chloroquine on SARS-Cov-2 replication?" and the corresponding rank and score in AskMe results. Table 2 shows the top ten AskMe results for the same query, using the default scoring values; the comparable LitCovid rankings are not available.[5] Documents in common in both top ten lists are in bold, showing a 40% overlap.

To evaluate the results of AskMe, we asked an expert to compare results from AskMe for several queries against results from LitCovid, Vespa, and CORD19.aws. The assessment shows that papers that the top 20 results from all of these search engines are very similar, with a few aberrations. For example, in one case where AskMe, Vespa, and CORD-19.aws rated the same documents at the top of their lists, LitCovid did not return that document among its top 20; and for one question AskMe's top few results were deemed highly relevant, while none of the other engines found these documents. Obviously, this is a preliminary assessment, but it gives us confidence that AskMe performs at the same level as other publication search engines–including engines such as CORD19.aws that use sophisticated machine learning techniques to filter results.

We will in the near future provide results of an evaluation using the TREC-COVID (CORD-19-based) benchmark (https://ir.nist.gov/covidSubmit/index.html) to test how AskMe performs on this labeled data, when compared to other search engines.

| LitCovid rank | Document ID | AskMe rank | AskMe score |
|---|---|---|---|
| 1 | **PMC7118659** | 3 | 7.809 |
| 2 | PMC7202847 | 76 | 5.197 |
| 3 | PMC7250542 | 115 | 4.175 |
| 4 | PMC7270792 | 61 | 6.001 |
| 5 | **PMC7232887** | 9 | 7.474 |
| 6 | **PMC7244425** | 1 | 9.906 |
| 7 | PMC7255230 | 98 | 4.804 |
| 8 | PMC7249615 | 128 | 3.64 |
| 9 | **PMC7108130** | 4 | 7.736 |
| 10 | PMC7275144 | 111 | 4.357 |

Table 1: LitCovid Query Results compared to AskMe

| AskMe rank | Document ID | AskMe score |
|---|---|---|
| 1 | PMC7244425 | 9.906 |
| 2 | PMC7127386 | 8.382 |
| 3 | PMC7128678 | 7.929 |
| 4 | **PMC7118659** | 7.799 |
| 5 | **PMC7108130** | 7.736 |
| 6 | PMC7314063 | 7.725 |
| 7 | PMC7217799 | 7.610 |
| 8 | **PMC7232887** | 7.474 |
| 9 | PMC7184359 | 7.370 |
| 10 | PMC7205718 | 7.365 |

Table 2: AskMe Query Results

## 6 Future Work and Conclusion

Our evaluation so far indicates that AskMe produces results on a par with LitCovid and several other publication search engines. As a front end to the LAPPS Grid coupled with the Grid's integration with annotation generators and editors, AskMe provides a unique, one-stop platform for data discovery, annotation, and text mining.

We are currently working on a number of performance and usability enhancements to the AskMe query engine, e.g., expanding scoring to other sections in full text articles and implementing a

---

[5]It is possible to search LitCovid results by title by searching 919 pages of results one-by-one, which was beyond the scope of our work at this point.

LAPPS Grid service to generate named entity annotations over CORD-19 documents for referral during the retrieval process. We are also adding nightly-generated embeddings from the CORD-19 data and integrating Tensorboard for their visualization. Our eventual goal is to migrate AskMe to ElasticSearch in order to enable integration with SemViz[6], thus providing an end-to-end system for query, search, and discovery over the CORD-19 data.

## Acknowledgements

This work was supported by NSF EAGER grant 1811402 and US Defense Advanced Research Projects Agency (DARPA), Contract W911NF-15-C-0238; Approved for Public Release, Distribution Unlimited. The views expressed are those of the authors and do not reflect the official policy or position of the Department of Defense or the U.S. Government. All remaining errors are, of course, those of the authors alone.

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
