# OpenReview forum: "AskMe: A LAPPS Grid-based NLP Query and Retrieval System for Covid-19 Literature"
_EMNLP/2020/Workshop/NLP-COVID — NLP-COVID19-EMNLP Poster_

### Official Review · AnonReviewer2 · 2020-09-17
**Good contribution but weak on evaluation**

**Rating:** 6
**Confidence:** 3

**Review:**

This work describes how the LAPPS Grid was augmented to support querying over the CORD-19 corpus. This extension makes available filtering features in Grid, as well as the suite of Grid NLP tools, for querying and accessing data in CORD-19. The authors suggest that Grid retrieval performs at a comparable level to other COVID search engines like Vespa or CORD19.aws, though no significant evaluation was performed.

It’s hard to gauge from the presented evaluation exactly how this system performs relative to other systems. The weakest part of the paper is the evaluation, which currently relies on a single expert evaluator comparing results from different search engines. The authors should consider evaluating on available CORD-19 retrieval datasets such as TREC-COVID (https://ir.nist.gov/covidSubmit/index.html).

---

### Official Review · AnonReviewer1 · 2020-09-22
**An interesting work with limited contribution and unclear evaluation conclusions.**

**Rating:** 5
**Confidence:** 3

**Review:**

The authors describe the AskMe system – a search engine utilizing the CORD-19 dataset to retrieve documents per natural language queries on the pandemic. The paper describes the engine in detail and provide a qualitative evaluation of its retrieval results to those of the LitCovid system. The paper is generally clear and well-written.

The weaknesses of this work lie in two main aspects: (1) scoring algorithm for relevant document retrieval, and (2) qualitative (and not entirely convincing) evaluation.

Re (1): Using only title and abstract for scoring a document is not considered sufficient for a retrieval task and would naturally render the proposed system inferior to others. Additionally, I appreciate the retrieval algorithm presented on page 3, but wonder how it compares to traditional (SOTA among them) IR approaches, including vector space, probabilistic and language models.

Re (2): It’s hard to draw any conclusions from (largely inconclusive) evaluation by a single expert, as presented on page 4. The authors could attempt, for example, the TREC-COVID (CORD-19-based) benchmark (https://ir.nist.gov/covidSubmit/index.html) and test how their engine performs on this labeled data, when compared to other search engines.

All in all, it seems like an interesting but not sufficiently mature work in its current form.

---

### Official Review · AnonReviewer3 · 2020-09-25
**Useful system that enables the search of latest literature with NLP-related features. However, the evaluation of the system is not convincing.**

**Rating:** 5
**Confidence:** 3

**Review:**

The paper describes AskMe system which is a query engine that searches COVID-19 literature and additionally offers further processing of the results using NLP tools.
It outlines the system briefly and makes a comparison with other related search engines, namely, LitCovid and iSearch.

While searching over the latest literature is indeed an important aspect, the search process is keyword-based (as opposed to vector-space-based) and limited to titles and abstracts.
In addition, as the authors acknowledge in the paper, only a preliminary evaluation is performed, and the result is not yet convincing.

Although the work in its current state is not yet mature, I do applaud the authors for building such a system that has a great potential to benefit medical researchers in the field.